# A Novel Hydrophilic, Antibacterial Chitosan-Based Coating Prepared by Ultrasonic Atomization Assisted LbL Assembly Technique

**DOI:** 10.3390/jfb14010043

**Published:** 2023-01-12

**Authors:** Xiaoyu Wang, Yuyang Zhou, Melissa Johnson, Cameron Milne, Sigen A, Yening Li, Wenxin Wang, Nan Zhang, Qian Xu

**Affiliations:** 1Charles Institute of Dermatology, School of Medicine, University College Dublin, D04 V1W8 Dublin, Ireland; 2Centre of Micro/Nano Manufacturing Technology (MNMT-Dublin), School of Mechanical and Materials Engineering, University College Dublin, D04 KW52 Dublin, Ireland; 3National Engineering Laboratory for Modern Silk, College of Textile and Clothing Engineering, Soochow University, Suzhou 215123, China

**Keywords:** chitosan, PLA, hydrophilic, antibacterial, coating

## Abstract

To explore the potential applicability of chitosan (CTS), we prepared aldehyde chitosan (CTS-CHO) with chitosan and sodium periodate via oxidation reaction and then a chitosan-based hydrophilic and antibacterial coating on the surface of poly (lactic acid) (PLA) film was developed and characterized. The oxidation degree was determined by Elemental analyser to be 12.53%, and a Fourier transform infrared spectroscopy was used to characterize the structure of CTS-CHO. It was evident that CTS-CHO is a biocompatible coating biomaterial with more than 80% cell viability obtained through the Live/Dead staining assay and the alamarBlue assay. The hydrophilic and antibacterial CTS-CHO coating on the PLA surface was prepared by ultrasonic atomization assisted LbL assembly technique due to Schiff’s base reaction within and between layers. The CTS-CHO coating had better hydrophilicity and transparency, a more definite industrialization potential, and higher antibacterial activity at experimental concentrations than the CTS coating. All of the results demonstrated that the ultrasonic atomization-assisted LbL assembly CTS-CHO coating is a promising alternative for improving hydrophilicity and antibacterial activity on the PLA surface. The functional groups of CTS-CHO could react with active components with amino groups via dynamic Schiff’s base reaction and provide the opportunity to create a drug releasing surface for biomedical applications.

## 1. Introduction

PLA is an aliphatic polyester and an environmentally friendly polymer. PLA is constructed from lactic acid (2-hydroxy propionic acid) building blocks [1,2]. Given its renewability, absorbability [1], biodegradability [3], and biocompatibility [4], it is widely applied in the biomedical field, such as absorbable surgical sutures [5], scaffolds [6,7,8], drug delivery systems [9,10], personal protective equipment [11,12], food packaging [13,14], and clothing fabrics and textiles [15]. PLA has been studied widely in the biomedical field during the global COVID-19 pandemic. However, the poor hydrophilicity limits its application in some fields, such as clothing, and medical devices [16], because the hydrophobic property leads to the absorption of non-specific protein and promotes bacterial adhesion [17,18]. Modification of the PLA surface can significantly improve the surface properties such as wettability, biocompatibility, and functionality of PLA [19,20,21].

Numerous surface modification strategies have been explored to improve the hydrophilicity of PLA, including physical, chemical, plasma, and radiation treatments. Among these methods, plasma-treatment is an effective tool for surface activation before grafting of bioactive components onto material surfaces [22,23,24,25,26,27,28]. For example, in Park et al.’s study [29], plasma treatment and hydrophilic acrylic acid grafting made hydrophilic functional groups successfully adapt on the surface of scaffolds [30]. Gutierrez-Villarreal et al. grafted N-vinylpyrrolidone onto PLA using benzophenone as the initiator and improved the hydrophilicity of PLA [31].

The LbL assembly technique allows coatings to be built based on the interactions of selected materials. Interactions exploited in the LbL method include covalent interactions, hydrogen bonds, and electrostatic attraction [32]. Ultrasonic atomization technology, which can convert the liquid into aerosols, has also been explored for developing multi-layered coatings [33]. The LbL assembly technique has been considered one of the most appropriate methods for preparing multilayer films incorporated with therapeutic molecules.

PLA’s antibacterial activity is one of the most important considerations, particularly in the fields of medical devices and food packaging. Among the various antibacterial materials which are commonly used on the PLA surface, natural polymers show great potential due to their excellent biocompatibility and degradability. Over the past decade, increasing research has gained insight into the antimicrobial activity of CTS, which is composed of N-acetyl-ᴅ-glucosamine and ᴅ-glucosamine units. In Munteanu et al.’s study, PLA films were coated with chitosan oil by coaxial electrospinning, combining the mechanical properties and biodegradability of PLA substrates with the antioxidant and antimicrobial properties of the chitosan–oil nano coatings [16]. However, the affinity of CTS to the surface of PLA is limited due to its poor solubility, high viscosity, and easy aggregation. Additionally, acidic solvents used to dissolve CTS are not suitable for industrial production. CTS-derived polymers fabricated through chemical functionalization have achieved the desired properties of polymer and coating, as well as industrial production. In particular, those CTS-derived polymers with antimicrobial activity have been extensively studied for the use on PLA surfaces for medical devices or food packaging [34,35]. However, the antibacterial coating on PLA surface prepared with aldehyde-chitosan (CTS-CHO) using ultrasonic atomization assisted LbL assembly technique has not been reported. 

In a recent study, the CTS-CHO was fabricated through chemical functionalization. CTS-CHO demonstrated the potential for industrial production of antibacterial coating due to its excellent properties, such as antibacterial activity, and solubility. Then an LbL assembly coating was prepared by ultrasonic atomization-assisted technique. The study process, including the preparation and characterization in this work is outlined in Figure 1. The characterization of CTS-CHO was performed using Fourier transform infrared spectroscopy (FT-IR) and an Exeter elemental analyser. The solubility, zeta-potential, stability, viscosity, and aggregation profiles of CTS and CTS-CHO were evaluated. Furthermore, the biocompatibility of CTS-CHO and CTS was assessed using Live/Dead staining assay and alamarBlue assay. After plasma treatment, active groups, such as hydroxyl and carboxyl groups, were generated on the PLA film surface. Then the bioactive inks prepared with CTS-CHO and CTS were sprayed on the surface of the PLA layer-by-layer through ultrasonic atomization, respectively. Due to the presence of aldehyde groups and amino groups in the structure of CTS-CHO, Schiff’s reaction occurred within and between layers. The transparency, antibacterial activity, and hydrophilicity of ultrasonic atomization-assisted LbL CTS-CHO coating and CTS coating on the PLA surface were then compared.

## 2. Materials and Methods

### 2.1. Materials

Chitosan with medium molecular weight, LB agar, and LB broth were purchased from Sigma-Aldrich (Dublin, Ireland). PLA film (0.5 mm) was purchased from Esun (Shenzhen, China). Sodium periodate and ethylene glycol were purchased from Thermo Fisher Scientific (Dublin, Ireland). Glacial acetic acid was supplied by Aladdin (Dublin, Ireland). Dialysis tubing (cut-off molecular weight: 8 kDa) were supplied by Spectrum Lab (Dublin, Ireland). *E. coli* and HEK 293 cells were supplied by ATCC. Cell culture medium was purchased from Invitrogen (Dublin, Ireland). Live/Dead staining kit was supplied by Biosciences (Dublin, Ireland) and an alamarBlue cell viability assay kit was purchased from Sigma-Aldrich (Dublin, Ireland), respectively. 

### 2.2. Fabrication of Antibacterial PLA Film

#### 2.2.1. Preparation of CTS-CHO

CTS-CHO was fabricated using chitosan (CTS) and sodium periodate via oxidation reaction, as reported previously [36]. 

#### 2.2.2. Bioactive Ink Formulation

The CTS-CHO freeze-dried sponge was dissolved in deionized water to the desired concentrations and vortexed for 30 min. 0.1 mol/L acetic acid solution was used to dissolve the CTS. Then the CTS solution was stirred for 24 h to ensure complete dissolution. 

#### 2.2.3. Preparation of Coatings 

The coating was prepared with a coating workstation (UAC120 Ultrasonic Atomizer System, Hangzhou, China). Prior to coating, PLA films were treated by plasma for 30~60 s under 30~80 W power. The plasma treatment was carried out in the oxygen atmosphere holding in a cylindrical chamber. The infusion rate was 0.5 mL/min, and the guide gas pressure was 0.01 MPa. After 20 times repeated coating, the films were placed in the open air to dry.

### 2.3. Characterizations

#### 2.3.1. CTS-CHO

Aldehyde chitosan was characterized by Fourier transform infrared spectrometry and an Exeter elemental analyser.

The infrared spectrograms of CTS and CTS-CHO were obtained by an ALPHA FT-IR spectrometer with an ATR accessory (Bruker, Dublin, Ireland). The samples were placed in the sampling area. The resolution was set to 4 cm^−1^. The spectrogram (32 scans) was recorded from 4000 cm^−1^ to 400 cm^−1^. The sample spectra were then subtracted from the background.

In this study, quantitative flash combustion was used for element analysis. After freeze-drying, CTS-CHO was kept in a desiccator before elemental analysis. An Exeter elemental analyser (CE440, Coventry, UK) was used to conduct this measurement. The oxidation degree (*F_ox_*) was calculated as below:(1)$$\frac{N}{C}=\frac{F_{A}+\left(1-F_{A}\right)\cdot \left(1-F_{ox}\right)}{2F_{ox}+6}$$
where *F_ox_* is the oxidation degree, *F_A_* is acetylated unit content, *C* is the carbon percentage, *N* is the nitrogen percentage.

#### 2.3.2. Bioactive Ink

The properties of bioactive inks were characterized by a zetasizer, viscometer, and microscope.

The viscosity of the ink was measured at 25 °C and 100 rpm with a rotational viscometer (AMETEK, Brookfield, MA, USA). The particle size of the inks was measured using a zetasizer before and after filtering with a 0.45 µm filter (Zetasizer Pro, Malvern, Ireland). Aggregation profiles of the ink were observed under a microscope (Olympus, Ireland). The zeta potential of the inks was tested using a zetasizer (Zetasizer Pro, Malvern, Ireland). The particle size of the ink was tested at predetermined time points using a zetasizer (Zetasizer Pro, Malvern, Ireland).

#### 2.3.3. Coating

The properties of the coatings were characterized by a digital camera, and contact angle goniometer, etc.

A digital camera was used to record the transparency of films, including the PLA film before and after plasma treatment, CTS and CTS-CHO coating on the PLA film surface. The measurement of the water contact angle was conducted using a Contact Angle Goniometer (Ossila, London, UK).

The antibacterial activity of plasma-treated PLA film, PLA films with CTS and CTS-CHO coating were tested according to a modified method based on ISO 22196. Briefly, a colony of *E. coli* was cultivated in LB broth overnight at 37 °C. The bacterial suspension was diluted with LB broth to a 0.5 McFarland concentration (~1.5 × 10^8^ CFU/mL). The bacterial solution was diluted to 1.5 × 10^5^ CFU/mL with LB broth to conduct the further experiment. 10 mL of bacterial solution was added into an ultrasonic atomizing sprayer. The PLA films with CTS and CTS-CHO coating were then cut into 5 cm × 5 cm squares and placed in the sterilized dishes. As a control, plasma treated PLA film without any coating was used and operated in the same way as described above. The bacterial solution was sprayed onto the Z-shaped films 3 times. Then 10 mL of LB broth was added to the dish with plasma-treated PLA film after inoculation. Then, 10-fold serial dilutions of the bacterial solution were performed with LB broth. 1 mL of each diluent was placed into two dishes, and 20 mL of LB agar was poured in. After gently swirling, the dishes were incubated for 2 days at 35 °C. After the bacterial solution was sprayed onto the films, 4 cm × 4 cm sterilized PE films were cut and placed onto the PLA films with coating and the plasma treated PLA films. 2 mL of sterilized water was dropped around the PLA films with coating and plasma treated PLA film to avoid bacterial solution drying. Then the dishes were incubated for 24 h at 37 °C. A pipette was used to remove the water from around the films. To recover the bacteria, 10 mL of LB broth was added to the dishes. 10-fold serial dilutions of the bacterial solution were performed with LB broth. 1 mL of each diluent was placed into two dishes and followed by 20 mL of LB agar. After gently swirling the dishes, they were incubated at 35 °C for 2 days. Then the antibacterial activity was calculated according to the equations in ISO 22196.

AlamarBlue assay and Live/Dead staining assay were used to evaluate the cytotoxicity of materials (CTS, CTS-CHO) against HEK293 cells. CTS and CTS-CHO were dissolved in sodium acetate buffer (25 mM, pH 5.2) and the stock solution was obtained after extraction for 24 h at 37 °C. Then 0.22 µm filters were used to filter the stock solutions for sterilization and the sterilized stock solutions were diluted with complete cell culture medium to various concentrations. 1 × 10^5^ cells/well of HEK 293 cells were inoculated in a 96-well microplate with complete medium at 37 °C (5% CO_2_). After overnight incubation, CTS or CTS-CHO solutions were used to replace the culture medium. After 24 h of incubation, 20 μL of alamarBlue reagent was added to each well. The fluorescence excitation wavelength was read at 570 nm by a microplate reader after 3 h of incubation. The negative control group consisted of cells grown in medium containing alamarBlue reagent. Live/Dead cell staining assay was performed according to the manufacturer’s protocol after 24 h incubation. 100 µL of staining solution was used to replace the culture medium at predetermined time points. Then DPBS was used to wash staining reagent away after 30 min incubation (25 °C). A fluorescence microscope was used to record the images.

## 3. Results and Discussion

### 3.1. Characterization of CTS-CHO

To confirm that the functionalization was successful, FT-IR was used to characterize the CTS and the CTS-CHO. The characteristic bands were shown in Figure 1. The stretching vibrations at 3283, 2924, 1647, and 1558 cm^−1^ are attributed to the O-H, C-H, C=O, N-H, and C-O-C linkages, which is consistent with previous literature reports. The band at 1725 cm^−1^ in CTS-CHO was the characteristic peak of aldehyde groups. Other bands, such as those at 3283, 2924, 1647, and 1558 cm^−1^, were not significantly altered in the oxidation reaction. These results were consistent with those reported in the literature [37,38,39,40]. Therefore, the C_2_-C_3_ linkage was cleaved in the GlcN unit, and the dialdehyde was formed in the structure of chitosan, as shown in Figure 1.

An elemental analyser was used to test the percentages of C, H, and N in the structure of CTS and CTS-CHO. The oxidation degree is defined as the dialdehyde percentage per 100 GlcN units. According to the literature, the oxidation degree was calculated with Equation (1). The oxidation degree was 12.53% (Appendix A).

### 3.2. Characterization of Bioactive Inks

To select the appropriate concentrations of CTS and CTS-CHO to conduct further study, the viscosity was measured by a rotational viscometer. A promising property of CTS-CHO is that its water solubility improved significantly compared to CTS. Presumably the hydration of aldehyde groups improved the solubility dramatically. Therefore, the CTS solution and CTS-CHO solution were prepared by dissolving materials in acetic acid solution and deionized water respectively. As shown in Table 1, the viscosity of the CTS solution increased with concentrations (from 1 mg/mL to 10 mg/mL). However, there was no obvious difference in the viscosity of the CTS-CHO solution (from 1 mg/mL to 10 mg/mL). The viscosity of the CTS-CHO solution was significantly lower than that of CTS solution at the same concentration. The appropriate ink viscosity is critical for ultrasonic atomization spray coating. High viscosity ink may cause ultrasonic atomizer pipe clogging, and uneven coating. Therefore, 10 mg/mL CTS-CHO solution and 1 mg/mL CTS solution were used in this study.

Chitosan can only be dissolved in acidic solutions at a pH below 6.0. However, according to the literature, a certain degree of aggregation exists even in a dissolved chitosan solution [41]. The CTS solutions consist of numerous intermolecular aggregates and molecularly dispersed polymer chains. Therefore, the uniformity of the deposited the CTS coating may be affected. Moreover, polysaccharide solutions usually exhibit a strong tendency to aggregate, including the CTS molecules in aqueous solvents. Ultracentrifugation, extensive filtering of solutions, and changes in solvent conditions such as temperature of exposure to acidic proteases, ionic strength, and pH are normally used to remove large aggregates. However, strong aggregation still cannot be prevented [29,42,43,44,45,46].

To observe the aggregation profiles of the CTS and CTS-CHO solution at a concentration of 1 mg/mL and 10 mg/mL, the particle size was measured using a zetasizer and microscope. As shown in Figure 2A–D, the obvious aggregation phenomenon of CTS was observed in the original solution and even the filtered solution. There are some particles over 20 µm in size after filtering with a 0.45 µm filter. The same results were observed for the particle size obtained from the zetasizer in Figure 2E,F. As the relative amount of amide groups decreased, the aggregation of CTS-CHO was significantly reduced, as shown in Figure 2G–L. After filtering, the majority of CTS-CHO particles were around 141 nm in size, but quite a few CTS particles were around 2 µm. Compared with CTS, the slight aggregation of CTS-CHO reduced the risk of pipe blockage and uneven coating.

The antibacterial activity of CTS depends on its positive charge. Positive charges bind to the negatively charged bacterial cell wall, causing damage to the cell wall and the alteration of the permeability of the cell membrane [47]. Chitosan then attaches to the DNA of bacterial cells and inhibits DNA replication, leading to bacteria death. Therefore, the zeta potential of the CTS solution and CTS-CHO solution was evaluated using a zetasizer. The results are shown in Appendix A. The potential of the 1 mg/mL CTS solution was 10.9 mv while the 10 mg/mL CTS-CHO solution has a potential of 28.8 mv. The results demonstrated that both CTS and CTS-CHO have positive charges, and thus possess potential antibacterial activity. These results were consistent with our previous report [36].

The stability of the inks also affects the ultrasonic atomization coating. As this study is the first report on the application of the CTS-CHO as an antibacterial coating material, the particle size of the solutions at predetermined time points was measured by zetasizer to observe the stability of CTS-CHO solution. The particle size was around 140 nm, and the PDI was around 0.25 (as shown in Figure 3). Both particle size and PDI showed no significant changes at 8 h, and only a slight increase at 24 h.

All of the results demonstrated that the bioactive inks prepared with CTS-CHO are a promising candidate for antibacterial coating material in the biomedical, food packaging and personal protection fields.

### 3.3. Characterization of Coatings

The photos of pristine PLA film, plasma treated PLA film, CTS coating and CTS-CHO coated PLA film were taken with a digital camera. The pristine PLA film and plasma treated PLA film showed high visible light transparency (see Figure 4A,B). Interestingly, with the 20 bilayers of the CTS-CHO coating, it is still possible to see the background beneath the film without any visible shelter or obscured area (see Figure 4D). However, the CTS coated PLA film showed poor transparency (see Figure 4C). The reduction of molecular weight and the improvement of solubility after the oxidation reaction enable the CTS-CHO coating to possess high transparency.

To evaluate the surface hydrophilicity, the water contact angle was measured with a contact angle goniometer. The results were shown in Figure 4E, where the contact angle of pristine was 77.9°, and that of plasma treated PLA film was reduced significantly to 14°. 83°. Water contact angles of CTS coating and CTS-CHO coating were 83.0 and 28.1°, respectively, which means that the hydrophilicity of the CTS-CHO coating was higher than that of the CTS coating.

A modified method based on ISO 22196 was used to evaluate the antibacterial activity in this study. As shown in Figure 5A–C, there were no bacterial colonies on the dishes after 48 h incubation for both the CTS-CHO coatings sprayed 10 times and 20 times. The antibacterial activity (R) of the CTS-CHO coating was around 4 in the experimental condition in this study. Therefore, when the CTS-CHO coating thickness reaches a certain level, the antibacterial activity will not increase with the spraying times and coating thickness. Figure 5D–F showed that the antibacterial activity of the CTS-CHO coating was significantly higher than that of the CTS coating. A large number of positively charged amino groups enable chitosan to have excellent antibacterial activity. After the oxidation reaction, the CTS−CHO still contains many amino groups which contribute to the antibacterial activity.

AlamarBlue measurement and Live/Dead staining assay were used to assess the cytotoxicity of the materials. The sodium acetate buffer (25 mM, pH = 5.2) was used to dissolve CTS and CTS-CHO to make sure that the experimental conditions were consistent. The cell viability results (>90%) indicated that the cytotoxicity of CTS-CHO didn’t increase after oxidation (Figure 6A,B).

Live/Dead assay was conducted after co-culturing CTS and CTS-CHO solution for 24 h. A fluorescence microscope was used to observe the outcomes. After 24 h following treatment with CTS solution and CTS−CHO solution, most HEK 293 cells were viable (Figure 6C–R). According to ISO 10993, the materials were non-cytotoxic.

## 4. Conclusions

A hydrophilic, antibacterial coating based on aldehyde chitosan was developed using nebulization assisted LbL assembly technology via Schiff’s base reaction. Aldehyde chitosan was fabricated via an oxidation reaction with chitosan and sodium periodate. According to the FT-IR spectra, aldehyde groups were introduced into chitosan successfully after the oxidation reaction. The oxidation degree of aldehyde chitosan in this study was 12.53%. The results of viscosity, zeta potential, particle size, and stability showed that aldehyde chitosan was a promising candidate for bioactive inks used in antibacterial coating fields. An ideal coating on the surface of the PLA film with hydrophilic, transparent properties and antibacterial activity was designed and characterized. The better transparency, lower water contact angle, higher antibacterial activity, and biocompatibility demonstrated that CTS-CHO coating will be an alternative to CTS as a material for antibacterial coating in the biomedical field, food packaging field, and personal protective field. Drug embedded layers can be assembled to realize drug loadings on the surface for sustained release. This study also represents one of the steps before further investigation for clinical use.

## Data Availability

The data presented in this study are available on request from the corresponding author.

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
