# Peer review of "A Novel Hydrophilic, Antibacterial Chitosan-Based Coating Prepared by Ultrasonic Atomization Assisted LbL Assembly Technique"

_jfb, 2023, doi:10.3390/jfb14010043_

Round 1

Reviewer 1 Report

This manuscript reports preparation and application of the aldehyde chitosan (CTS-CHO). The work is feasible. But the manuscript still needs improvement.

1. The scope of the title is too large. It is suggested to narrow the scope. E.g., A novel hydrophilic, antibacterial CTS-CHO coating based on ultrasonic atomization assisted LbL assembly technique?

2. I don't think the innovation of this research has been highlighted in the Introduction, please improve.

3. Under Results and discussion, “In the recent study, we designed a hydrophilic and antibacterial coating on the surface of PLA film with aldehyde chitosan. Aldehyde chitosan was fabricated with chitosan and sodium periodate via oxidation reaction following the procedure we reported previously [36]. Then the coating was prepared based on ultrasonic nebulization assisted LbL assembly technique.” Why repeat this? This was explained in the experimental section, it's unnecessary.

4. For comparison, the FTIR of CTS before modification should be given.

5. There are drawing non-standard problems. Please improve. E.g., Unit of coordinates in Fig.1, [%]? cm-1?

Author Response

This manuscript reports preparation and application of the aldehyde chitosan (CTS-CHO). The work is feasible. But the manuscript still needs improvement.

Re: We really appreciate the reviewer’s comments which helped us to improve our research work. In order to address the raised concerns, we have listed our point-by-point response below.

  1. The scope of the title is too large. It is suggested to narrow the scope. E.g., A novel hydrophilic, antibacterial CTS-CHO coating based on ultrasonic atomization assisted LbL assembly technique?

Re: We appreciate the reviewer’s suggestion. The title of this manuscript has been changed to “A novel hydrophilic, antibacterial chitosan-based coating prepared by ultrasonic atomization assisted LbL assembly technique”.

  1. I don't think the innovation of this research has been highlighted in the Introduction, please improve.

We thank the reviewer’s suggestion. In the introduction section, we revised the manuscript and supplemented some information on the novelty of our research. “In some fields, especially medical devices and food packaging, the antibacterial activity of PLA is one of the foremost considerations. Among the various antibacterial materials which are commonly used on PLA surface, natural polymers show great potential due to the excellent biocompatibility and degradability. Over the past decade, increasing research has gained insight into the antimicrobial activity of chitosan (CTS), which consists of N-acetyl-ᴅ-glucosamine and ᴅ-glucosamine units. In Munteanu et al’s study, PLA films were coated with chitosan oil by coaxial electrospinning, combining the mechanical properties and biodegradability of PLA substrates with the antioxidant and antimicrobial properties of the chitosan–oil nano coatings [16]. However, the affinity of CTS to the surface of PLA is limited due to its poor solubility, high viscosity, and easy aggregation. Additionally, acidic solvents used to dissolve CTS are not suitable for industrial production. CTS-derived polymers fabricated through chemical functionalization have achieved the desired properties of polymer and coating, as well as industrial production. In particular, those CTS-derived polymers with antimicrobial activity have been extensively studied for the use on PLA surfaces for medical devices or food packaging [34, 35]. However, the antibacterial coating on PLA surface prepared with aldehyde-chitosan (CTS-CHO) using ultrasonic atomization assisted LbL assembly technique has not been reported.  (line 63~80).”

  1. Under Results and discussion, “In the recent study, we designed a hydrophilic and antibacterial coating on the surface of PLA film with aldehyde chitosan. Aldehyde chitosan was fabricated with chitosan and sodium periodate via oxidation reaction following the procedure we reported previously [36]. Then the coating was prepared based on ultrasonic nebulization assisted LbL assembly technique.” Why repeat this? This was explained in the experimental section, it's unnecessary.

Re: We appreciate the reviewer bringing this to our attention. We revised the manuscript and deleted the repeated words.

  1. For comparison, the FTIR of CTS before modification should be given.

Re: We thank the reviewer for this suggestion. The spectrum of CTS has been added to the manuscript (as shown in Figure 1).

  1. There are drawing non-standard problems. Please improve. E.g., Unit of coordinates in Fig.1, [%]? cm-1?

Re: Thanks for this suggestion. Figure 1 has been revised.

Reviewer 2 Report

The present work studies a novel hydrophilic antibacterial coating based on the LbL assembly technique assisted by ultrasonic atomization. The work has excellent potential for application in the biomedical area. Basically, due to the renewability, absorption capacity, biodegradability, and biocompatibility of chitosan aldehyde and also for the simplicity of the technique used for the growth of the cover. Overall, the manuscript was well-written and designed. However, some improvements are needed.

Suggestions:

i) It is necessary to add more details of the treatment performed with plasma, as it was not described which reactor was used, nor which plasma parameters were used;

ii) In a manuscript, it is essential to number the lines for easier correction;

iii) It is necessary to add references in the following paragraphs:

"But according to the literature [ref], a certain degree of aggregation exists even in dissolved chitosan solution."

"The antibacterial activity of chitosan depends on the positive charges. Positive charges bind to the negatively charged bacterial cell wall, causing the damage of the cell wall and the permeability of the cell membrane was altered. [ref]"

Author Response

The present work studies a novel hydrophilic antibacterial coating based on the LbL assembly technique assisted by ultrasonic atomization. The work has excellent potential for application in the biomedical area. Basically, due to the renewability, absorption capacity, biodegradability, and biocompatibility of chitosan aldehyde and also for the simplicity of the technique used for the growth of the cover. Overall, the manuscript was well-written and designed. However, some improvements are needed.

We really appreciate the reviewer’s valuable suggestions which helped us to improve our manuscript. Here, we have listed our point-by-point response below.

Suggestions:

  1. i) It is necessary to add more details of the treatment performed with plasma, as it was not described which reactor was used, nor which plasma parameters were used;

Thanks for this suggestion. We have revised the manuscript and supplemented some information. “   2.2.3. Ultrasonic atomization spray coating (UASC)  UASC was carried out on a customized surface coating workstation equipped with the UAC120 Ultrasonic Atomizer System (Cheersonic Ultrasonic Co.). Before UASC, PLA films were rinsed with anhydrous alcohol to remove the dust and oil during film production and subjected to plasma treatment using under 30~80W power for 30~60s. The plasma treatment was carried out in the oxygen atmosphere holding in a cylindrical chamber. UASC was implemented strip by strip at an infusion rate of 0.5 mL/min under 0.01 MPa guide gas pressure repeated 20 times. The resultant films were dried in open air.”

  1. ii) In a manuscript, it is essential to number the lines for easier correction;

Thanks for the reviewer’s suggestion. We have added line numbers in the manuscript.

iii) It is necessary to add references in the following paragraphs:

"But according to the literature [ref], a certain degree of aggregation exists even in dissolved chitosan solution."

"The antibacterial activity of chitosan depends on the positive charges. Positive charges bind to the negatively charged bacterial cell wall, causing the damage of the cell wall and the permeability of the cell membrane was altered. [ref]"

Re: Thanks for this suggestion. We have cited two references in the manuscript.

[41] Philippova, O.E., Korchagina, E.V. Chitosan and its hydrophobic derivatives: Preparation and aggregation in dilute aqueous solutions. Polym. Sci. Ser 2012, A 54, 552–572.

[47] Yilmaz Atay H. Antibacterial Activity of Chitosan-Based Systems. Functional Chitosan, 2020, Mar 6:457–89.